# COVID-19 Vaccination Personas in Yemen: Insights from Three Rounds of a Cross-Sectional Survey

**DOI:** 10.3390/vaccines11071272

**Published:** 2023-07-21

**Authors:** Zlatko Nikoloski, Dennis Chimenya, Abdullah Alshehari, Hauwa Hassan, Robert Bain, Leonardo Menchini, Amaya Gillespie

**Affiliations:** 1London School of Economics and Political Science, Houghton Street, London WC2A 2AE, UK; 2UNICEF Yemen, Sana’a P.O. Box 725, Yemen; dchimenya@unicef.org (D.C.); aalshehari@unicef.org (A.A.); hauhassan@unicef.org (H.H.); 3UNICEF Middle East and North Africa Regional Office, Amman 11821, Jordan; rbain@unicef.org (R.B.); lmenchini@unicef.org (L.M.); agillespie@unicef.org (A.G.)

**Keywords:** COVID-19, vaccination, Yemen

## Abstract

We used three rounds of a repeated cross-sectional survey on COVID-19 vaccination conducted throughout the entire territory of Yemen to: (i) describe the demographic and socio-economic characteristics associated with willingness to be vaccinated; (ii) analyse the link between beliefs associated with COVID-19 vaccines and willingness to be vaccinated; and (iii) analyse the potential platforms that could be used to target vaccine hesitancy and improve vaccine coverage in Yemen. Over two-thirds of respondents were either unwilling or unsure about vaccination across the three rounds. We found that gender, age, and educational attainment were significant correlates of vaccination status. Respondents with better knowledge about the virus and with greater confidence in the capacity of the authorities (and their own) to deal with the virus were more likely to be willing to be vaccinated. Consistent with the health belief model, practising one (or more) COVID-19 preventative measures was associated with a higher willingness to get a COVID-19 vaccination. Respondents with more positive views towards COVID-19 vaccines were also more likely to be willing to be vaccinated. By contrast, respondents who believed that vaccines are associated with significant side effects were more likely to refuse vaccination. Finally, those who relied on community leaders/healthcare workers as a trusted channel for obtaining COVID-19-related information were more likely to be willing to be vaccinated. Strengthening the information about the COVID-19 vaccination (safety, effectiveness, side effects) and communicating it through community leaders/healthcare workers could help increase the COVID-19 vaccine coverage in Yemen.

## 1. Introduction

Yemen has been struck by a devastating civil war that has significantly impacted the country’s overall quality of life since 2011. The war has resulted in a significant number of deaths and many injuries, with many more forced to flee their homes due to the protracted hostilities. Reports of grave children’s rights violations and gender-based violence have increased [1]. In 2021, 20.7 million people (66% of the population) were estimated to be in need of humanitarian assistance. It was estimated that 16.2 million people (more than half of the population) were hungry in 2021, and over 15.4 million people (around half the population) were in need of support to access water and sanitation. Only about half (51%) of the healthcare facilities in Yemen are fully functional, and the health worker density is only 10 per 10,000 population, compared to the WHO benchmark of 22 per 10,000 [2]. About 20.1 million Yemenis (62%) are in need of health assistance. At least one child dies every ten minutes in Yemen due to preventable diseases. Furthermore, there are ongoing challenges, such as the lack of salaries for health personnel and difficulties importing medicines and other critical supplies [1].

Against this difficult background, the first COVID-19 case was registered in Yemen in April 2020, followed by warnings of a potentially catastrophic outbreak [3]. Since April 2020, the virus spread across the country, although the total number of infections and deaths due to COVID-19 was difficult to ascertain, given the poor capacity of the Yemeni healthcare system [4]. Nevertheless, a recent examination of burial activities based on satellite imagery in the governorate of Aden during the pandemic revealed that COVID-19 had had a significant, underreported impact [5].

The immunisation programme was launched on 20 April 2021 (covering 13 of the 21 governorates) [6]. Yemen received 360,000 doses of AstraZeneca COVID-19 vaccinations as the first batch under the COVAX programme, according to the WHO Yemen Situation Report for March 2021. However, as of September 2022, Yemen has one of the lowest COVID-19 vaccination coverage rates globally, with about 5% of adults in Yemen having received at least one dose of the COVID-19 vaccine [7]. Various barriers have prevented the country from increasing COVID-19 vaccination coverage, including pre-existing barriers such as vaccine hesitancy, lack of adequate supplies of vaccines in Yemen, and political instability [3]. The existing literature suggests that these barriers were amplified during the COVID-19 pandemic.

### Studies

A study by Bitar et al. [8] relied on a sample of 484 participants and focused on two major questions: the main characteristics of misinformation and the main characteristics of vaccination hesitancy or rejection (the study was carried out before the immunisation campaign in Yemen had begun). University educated, higher income, employed, males living in urban areas were associated with lower misinformation about vaccination in general. In the same study, the acceptance rate for vaccination was 61% for free vaccines, and it decreased to 43% if participants had to purchase it. Females, respondents with lower monthly income, and those who believed that pharmaceutical companies made the virus for financial gains were more likely to reject the COVID-19 vaccination [8]. While beliefs were the main focus of the study by Bitar and colleagues [8], Noushad et al. [9] argued that severe shortage and lack of access to vaccines drove the low vaccination rates in the country. According to their study (conducted via WhatsApp survey to 5329 participants), over half of the respondents were willing to be vaccinated [9]. Finally, in a study by Bin Ghouth et al. [10], beliefs about the vaccine’s lack of safety and bad quality significantly contributed to a lower willingness to be vaccinated.

While there is some information on vaccination willingness and hesitancy in Yemen, the information is incomplete, as outlined above. To date, most of the existing evidence on vaccination uptake is based on one-off, small-scale surveys conducted using convenience sampling and are not representative of the population of Yemen. Against this background, this research paper aims to provide better understanding of the main correlates of vaccination intention and vaccination hesitancy, relying on a repeated cross-sectional survey conducted across Yemen. More specifically, the objective of this research paper is to describe three vaccination “personas” (willing, unwilling to be vaccinated, and unsure) in terms of (i) demographic and other individual characteristics (including knowledge and exposure to COVID-19); (ii) their main vaccination related beliefs (e.g., safety, side effects); and (iii) their preferred channels for reaching communities (e.g., community leaders, social media).

## 2. Methodology

### 2.1. Survey Instrument

We used three rounds of a survey titled “Rapid assessment of knowledge, attitudes and practices related to COVID-19” in Yemen. The survey was implemented in five rounds; however, this paper focuses on the last three rounds, where the questions on vaccination intention were asked (March 2021, August/September 2021 and April 2022).

There were about 1400 respondents per round across the entire country. The sample size was determined based on several factors, a population size of 14 million people (population at age > 17), a confidence level of 95%, and a margin of error of approximately 2.5%. The formula for calculating sample size:Sample size= (z^2^ xp (1 − p)/e^2^)/1 + (z^2^ xp (1 − p)/e^2^ N)

N—population size, e—Margin of error (percentage in decimal form), z—z-score.

The three rounds of the survey followed a repeated cross-section format (rather than a longitudinal survey format); thus, the same individuals did not appear in all three rounds of the survey.

The survey was administered over the phone in the south of the country and face-to-face in the northern part. In the north of the country, for the selection of the enumeration areas, governorates were identified to serve as primary sampling units (PSUs); based on this, governorates were implicitly stratified to allow for a random selection of clusters while considering the ease of access during the selection (e.g., not a conflict-affected zone, no restrictions from authorities). In turn, a simple random selection was applied for selection to be interviewed in each governorate. By contrast, the interviews in the south were carried out over the phone. More specifically, random numbers were selected from a dataset of phone numbers in the south (noting that this method impacts upon the representativeness of the sample in the south). The application of these different data collection methods did not significantly impact the response rate across the country. In other words, the number of interviews conducted are equal to the specified sample size in each round, both in the south and in the north. 

The objective of the survey was to: (i) describe the demographic and socio-economic characteristics associated with a willingness to be vaccinated; (ii) analyse the link between beliefs associated with COVID-19 vaccines and willingness to be vaccinated; and (iii) analyse the potential platforms that could be used in order to target vaccine hesitancy and improve vaccine coverage in Yemen. The survey instrument included items related to (i) knowledge of symptoms, transmission, and prevention; (ii) peoples’ sources of information; (iii) risk perception; (iv) information needs of respondents; (v) COVID-19-related stigma; and (vi) hesitancy or acceptance of COVID-19 vaccine. The questionnaire used for the data collection underwent a thorough review process, with input from several partners and counterparts, including the World Health Organisation as well as members of various United Nations and government coordination and decision-making bodies such as the COVID-19 task force and the risk communication and community engagement working group. Additionally, the questionnaire was pre-tested with selected participants to ensure clarity and relevance. The questionnaire is available upon request.

### 2.2. Statistical Analysis

We adopted a descriptive analysis of the main characteristics of three vaccination personas: (a) those willing; (b) unsure if they wanted to be vaccinated, and (c) those unwilling. In order to distil the three personas, we relied on the following question from the survey: “Would you be willing to get the COVID-19 vaccine when one becomes available in Yemen?”. Furthermore, the characteristics of the three personas were grouped into three major groups: (i) socio-demographic characteristics (e.g., age, gender, occupation), practising public health and social measures (PHSM), risk perception and trust in authorities); (ii) a second group relating to attitudes and beliefs towards the COVID-19 vaccines (e.g., beliefs in the vaccine safety and side effects); (iii) the final group of characteristics corresponding to the preferred channels for reaching different personas. As outlined above, in order to understand the characteristics of the different vaccination categories, we conducted a descriptive analysis, coupled with chi2 test of the difference between categorical variables. In carrying out the analysis, we focussed on the last round of the survey (round 5) and provide the analysis of the previous two rounds in Appendix A of the paper.

## 3. Results

### 3.1. Overall Description of the Sample

Table 1 provides a socio-demographic snapshot of the sample, across the three rounds. About one-third of respondents had completed secondary education and another quarter had completed some college degree, and roughly four-fifths of respondents were less than 50 years of age. Only a fraction of the sample had no or very little formal education. In round 5, about 7% of respondents could not read or write, while 15.6% had basic reading and writing skills. There were more males than females in the sample; more specifically, by the fifth round of the survey, about two-thirds of the sample consisted of males. Furthermore, the sample was almost equally split between the professions included in the study (educators, housewives, students, and office workers).

Between round 3 (March, 2021) and round 5 (April, 2022), the share of respondents who believed they could become infected with COVID-19 had increased. By the fifth round of the survey data, almost half of respondents stated they felt at risk of being infected by the virus. Over time, there was an increase in confidence regarding COVID-19 information provided by the authorities, coinciding with enhanced management of COVID-19. By the fifth round, over half of respondents reported confidence or total confidence in official COVID-19 information from the authorities. However, 14.6% of respondents still had no confidence and might resist authorities’ appeals for vaccine uptake. In addition, the reopening of the country, coupled with the relaxing of some of the stringent measures aimed at containing the virus, resulted in a reduction in the share of respondents practising various public health and social measures (PHSM). More specifically, by the fifth round, only about four percent of respondents practised social distancing (over the last four weeks), while about a third wore a mask in public, whereas handwashing seemed to be a more embedded habit with close to half of respondents (42.6%) still reporting washing their hands regularly with soap and warm water.

Figure 1 provides a summary of vaccination intention over time. There are a few important findings that stem from this analysis. While initially, the share of respondents not willing to be vaccinated had decreased (between rounds 3 and 4), there was very little change between rounds 4 and 5. More specifically, roughly 41% of respondents stated that they were not willing to receive the COVID-19 vaccination when it became available. Second, between rounds 3 and 4, there was an increase in the share of people willing to be vaccinated; however, it had reduced between rounds 4 and 5, at the expense of respondents who were not sure/undecided. By round 5, 28.2% of respondents were willing to be vaccinated, while 30.7% reported that they were unsure.

Figure 2 depicts the practice of various PHSM over time. There are a few major findings that stem from this chart. First, as the pandemic ebbed, the authorities were less stringent regarding enforcement of various measures to stop the transmission of the virus. Indeed, as the chart shows, the share of people practising PHSM over the last four weeks is roughly half compared to the share of the respondents practising the same type of PHSM in the previous ten months. In addition, there are visible differences in the prevalence of different PHSM. Handwashing (albeit measured only in rounds 4 and 5) is the most prevalent and sustained type of PHSM. For example, in round 5, just forty percent of respondents had practised handwashing in the last four weeks. Noting that handwashing pre-dated COVID-19 and is relevant well beyond COVID-19, its endurance over other PHSM was understandable. By contrast, about one-third of respondents reported mask-wearing (face covering). The rest of the PHSM were practised by a lower share of respondents, which had drastically dropped over time. This was particularly the case with measures such as not attending the mosque and avoiding social gatherings.

### 3.2. Vaccination Personas

#### 3.2.1. Persona 1: Willing to Be Vaccinated

There is some scant evidence that those willing to get vaccinated were slightly younger (Table 2), although the relationship between age and willingness to vaccinate is statistically insignificant. Furthermore, about a third of those with college degrees and close to half of respondents with higher degrees tended to be willing to be vaccinated. Consistent with the established notion from other countries and studies of other health practices, men were more likely than women to be willing to receive a COVID-19 vaccination. About a third of those who felt at risk of becoming infected with the virus were willing to receive at least one dose of the vaccine. Table 2 also provides some evidence that this vaccination persona tended also to adhere to public health and social measures (PHSM). For example, more than a third of those who practised social distancing were willing to receive the COVID-19 vaccine. Similarly high was the share of these respondents who stayed away from the mosque and were willing to receive a vaccination.

Table 3 summarises the analysis of vaccination status and knowledge regarding COVID-19. There are a few conclusions that stem from the table. First, the willingness to be vaccinated increased as knowledge about protecting oneself from the virus increased. More specifically, 40.6% of those with excellent knowledge about how to protect themselves were willing to be vaccinated. Similarly, willingness to be vaccinated increased as trust in the official information from authorities and their ability to deal with the virus increased. In addition, willingness to be vaccinated is a function of risk perception of the dangers of the virus. For example, 40.3% of respondents who thought the virus was dangerous were willing to be vaccinated.

We next turned to the link between vaccination status and beliefs about COVID-19 vaccines (Table 4). Consistent with the existing research, positive beliefs about the vaccine are associated with a higher willingness to be vaccinated. Nearly half (48.2%) of respondents who thought that the vaccine is effective were willing to be vaccinated. Similar findings emerged when considering beliefs about side effects. The results from the previous two rounds are reported in Appendix A, Table A3, and they were consistent with the findings emerging from round 5.

Various sources of information could be used as a vehicle to increase vaccine acceptance and, thus, vaccine uptake. This, however, depends on what type of information source is most trusted vis-à-vis COVID-19 vaccines. Against this background, we next turned to the link between vaccination status and the most trusted source of information (Table 5). Half (50%) of respondents listing community leaders as the most trusted COVID-19 information source were willing to be vaccinated (Table 5). Similar findings emerged from the previous two rounds (Appendix A, Table A4). In addition, in some of the previous rounds (e.g., round 3) we also found evidence that those who listed community healthcare workers as a trusted source of information were more likely to be willing to receive a COVID-19 vaccination. This persona tends to trust communication materials and community leaders more than other personas trust these sources of information.

#### 3.2.2. Persona 2: Not Vaccinated and Undecided

As in the case above, here as well, age, gender, and education were the main correlates of this persona (not vaccinated and undecided). A large share of the unemployed (38%) were undecided regarding a possible vaccination, suggesting a link to employer encouragement being a strong incentive for vaccination. About a third of those with no opinion regarding potential infection with the virus were undecided regarding obtaining a vaccine. Furthermore, no discernible link emerged between practising PHSM and being undecided about potentially obtaining a COVID-19 vaccination. About a quarter of those who believed that the vaccine is effective were undecided regarding taking it (slightly lower compared to those who did not think that there were serious side effects if/when taking the vaccine). This persona appeared to draw information from a wide range of sources, which may be contradictory.

#### 3.2.3. Persona 3: Not Willing to Get Vaccinated

As with the persona above, here as well, we found some evidence that this vaccination persona was older than the other categories. In addition, less educated by a significant margin. About two-thirds of respondents who could not read and write were not willing to get vaccinated. About half of women were unwilling to obtain a COVID-19 vaccine (about 15 percentage points higher than men). Almost two-thirds (63.5%) of respondents who stated that they did not believe they were likely to get infected with the virus were also unwilling to be vaccinated. Table 2 also provides the results of the link between vaccination status and practising different PHSM (wearing a mask in public, washing hands, keeping physical distance, and staying away from crowds/the mosque). The question on the PHSM practice was asked in reference to two time periods: 10 months ago and four weeks ago. The results of this analysis were unequivocal: those who did not practice PHSM were also less likely to be willing to be vaccinated. For example, 48.3% of respondents who claimed they did not wear a mask in public were unwilling to be vaccinated.

This vaccination persona was less knowledgeable about the COVID-19 virus (Table 3). For example, 81.4% of those with no knowledge were unwilling to obtain the vaccine. By the same token, this persona tended to believe that the virus is not dangerous. More specifically, 72.5% of those claiming the virus is not dangerous were unwilling to be vaccinated. Furthermore, this vaccination persona held negative attitudes and beliefs towards the vaccines. For example, about half (52%) of respondents who did not think that the vaccine is effective were unwilling to be vaccinated (Table 4). Finally, this group of people tended to trust their family and friends more than other personas for information regarding COVID-19.

As a complementary analysis, we also conducted the standard logit modelling analysis, where the three vaccination personas appeared as dependent variables in three separate models. The explanatory variables were grouped into three major groups: (i) socio-demographic variables (e.g., age, gender); (ii) practising some of the most common public health and social measures (e.g., wearing a mask, washing hands); and (iii) beliefs about the COVID-19 vaccines (e.g., effectiveness, side effects). The results are reported as Appendix A tables (Table A4, Table A5 and Table A6). The analysis supports the findings from the descriptive statistics; more specifically, certain demographic variables (e.g., gender) and variables capturing beliefs about COVID-19 vaccines explained the decision to obtain a COVID-19 vaccination.

In order to capture the PHSM/vaccination status nexus over time, we pooled the three waves together and used the three vaccination personas as dependent variables in three separate bivariate logit models (where the variables capturing different PHSM were used as independent variables). We repeated the analysis twice, first using PHSM practised over the last ten months and then over the last four weeks. The models also controlled for the survey wave (i.e., taking into account any temporal changes occurring over the three different waves). The findings (reported in Appendix A, Table A7 and Table A8) were unequivocal: those more willing to be vaccinated were also more willing to adhere to various PHSM (both over the last ten months as well as over the last four weeks).

## 4. Discussion

To the best of our knowledge, this is the first comprehensive attempt to describe various vaccination personas in Yemen, relying on a sample covering the entire country and spanning three points in time. In that respect, there are a few interesting findings that emerge from this study. First, our findings on the socio-demographic characteristics of vaccination willingness are consistent with the existing evidence. A recent paper using two waves of repeated cross-sectional surveys from the Middle East, North Africa, and Eastern Mediterranean region [11], for example, found that men, on average, were more likely to be vaccinated and to be willing to be vaccinated once vaccines were available to them. The same study also posits that men may be also advantaged by their higher level of mobility than women in parts of the region, and their higher engagement in formal employment, which may offer additional incentives for vaccination. The same study showed that women were disproportionately affected by misinformation about fertility, which also seemed to affect their willingness to be vaccinated. In addition, it has been argued that women are more likely to embrace conspiracy theories about the virus [12]. Other potential factors that can contribute to higher rates of vaccine hesitancy among females include the higher levels of fear of injections or side effects and the observation that the disease is more deadly in males [12]. Furthermore, in countries where men have greater access to healthcare services and the means to pay for vaccination than women, men may be more interested in the COVID-19 vaccination [13,14]. 

A study by Bitar et al. [7], also found that men were more likely to be willing to be vaccinated, while women were more likely to reject the vaccine. That study also finds that those with lower income are likely to reject the vaccines. While in our study, we did not have a variable capturing income, our variable on education attainment could be considered as a proxy for socio-economic status.

We also found that respondents who were practising some forms of preventative measures (e.g., wearing a mask, washing hands, practising social distancing) were more likely to be willing to obtain a vaccination. This finding supports the general health motivation construct in the health belief model [15], and aligns with social identity theory [16], which suggests that people who practise one health behaviour (such as vaccination) are more likely to practise others, such as PHSM in relation to the containment of COVID-19. Some of these associations were explored in a recent paper involving two rounds of repeated cross-sectional data on 14,000 respondents from the wider MENA region [11].

One of our principal findings relates to the link between vaccine beliefs and willingness to be vaccinated. To date, a large body of evidence stemming from the Middle East, North Africa, and Eastern Mediterranean region has also documented the link between vaccine beliefs and vaccination status. A study about vaccination among healthcare workers in Egypt, for example, found that the reasons for vaccine acceptance revolved around safety and effectiveness, while fear of side effects was the main reason for vaccine hesitancy [17]. Concerns about safety as well as a general lack of trust in the vaccines, were the main reason for vaccine hesitancy among healthcare workers in Sudan and Iraq [18,19]. Lack of trust in vaccine effectiveness and fear of side effects were the also main reasons for refusing to be vaccinated among the general population [17,20,21,22], while the belief in the effectiveness and benefits associated with the COVID-19 vaccination were the main reasons for vaccine acceptance [20,23].

These findings need to be interpreted within the broader context of the political situation in Yemen, which affected the availability of accurate information and vaccination services (including the availability of vaccines), particularly in the northern DFA (de facto authority)-controlled provinces. Across Yemen, a variety of misinformation about COVID-19 immunisation has taken root. The most frequently stated reasons for poor vaccination uptake by key informants in a study by Bin Ghouth and Al-Kaldy [9] were comparable to the findings of a sub-national survey carried out in early 2021 [24]. Some participants in that study saw the vaccination as a planned “scheme” that posed a danger to their health. Some individuals felt that the vaccination would cause death over time rather than instantly. Some claimed that the vaccine effort is a plot to create Muslim infertility [25]. Others said that the West was supplying Yemen with inadequate vaccinations [26]. People in the northern regions, on the other hand, did not see COVID-19 as a danger [24].

Finally, we found that respondents using certain sources of information (e.g., community leaders and volunteers) were more likely to be willing to be vaccinated. Compared to the regional average, trust in health workers is lower in Yemen, which can reasonably be expected to have an impact on vaccine uptake; the research in the area of vaccine demand generation has distilled two approaches. The first, more passive one, has relied on the use of mass media (TV and radio) and printed materials (banners, leaflets, posters) [27]. The second approach involved deeper face-to-face engagement with households and individual caregivers—often by trained volunteers from the community using interpersonal communication and behaviour change approaches. The success of this approach relies on extensive efforts by the community outreach workers to directly interact with the community as well as with individual caregivers. Even though the second approach is more labour intensive (and more expensive), it may also yield higher returns per contact when it comes to vaccination uptake, especially given the lower trust in health workers in Yemen.

There are some limitations associated with this research. First, the analysis is descriptive and only explores the correlation between vaccination status and the variables of interest. Correlations may be confounded by other observed and unobserved variables. In that respect, we cannot infer any direct causal links by using this methodological approach. Second, some questions changed over the course of the five rounds (e.g., additional categories were added to the most trusted source of information question), which may have some implications on the overall responses collected through this question. As the estimation and projection of demographic data in Yemen is of poor quality, the survey did not develop survey weights. More specifically, the results were not weighted for survey weights to address the representativeness of the sample. These limitations notwithstanding, there are some broad conclusions that stem from this research. First, we found that gender and socio-demographic status (e.g., education attainment) were significant correlates of vaccination status, consistent with existing knowledge. Second, respondents with better knowledge about the virus and with better confidence in authorities’ (and their own) capacity to deal with the virus were more likely to be willing to be vaccinated. Consistent with the health belief model, practising one (or more) preventative measures in relation to COVID-19 was associated with a higher willingness to get a COVID-19 vaccination. In addition, beliefs around the COVID-19 vaccines were also linked to willingness (or lack of willingness) to obtain a vaccination. Finally, those who relied on community leaders/healthcare workers as trusted sources of COVID-19-related information were more willing to be vaccinated.

Finally, there are some broad policy recommendations that stem from this research effort. Any focus on individual motivation for vaccination relies on the basic requirement that adequate vaccination services are made available to all communities. That said, outreach to communities and a localised focus on the needs of those who are undecided about vaccination can be effective in increasing uptake, thereby also increasing the social norm around being vaccinated. Supplying them with information about the COVID-19 vaccines (e.g., safety, effectiveness, and side effects) and access to trusted and skilled health workers could mitigate fears and increase confidence in the vaccines. Identifying vaccination champions among families/communities could further allay some of the fears associated with vaccines (e.g., fears of side effects). Religious leaders and other community leaders (including females) can have a strong influence on communities in Yemen, both positively and negatively—and should be considered key partners, especially in terms of understanding and addressing the needs of local communities.

## Figures and Tables

**Figure 1 vaccines-11-01272-f001:**
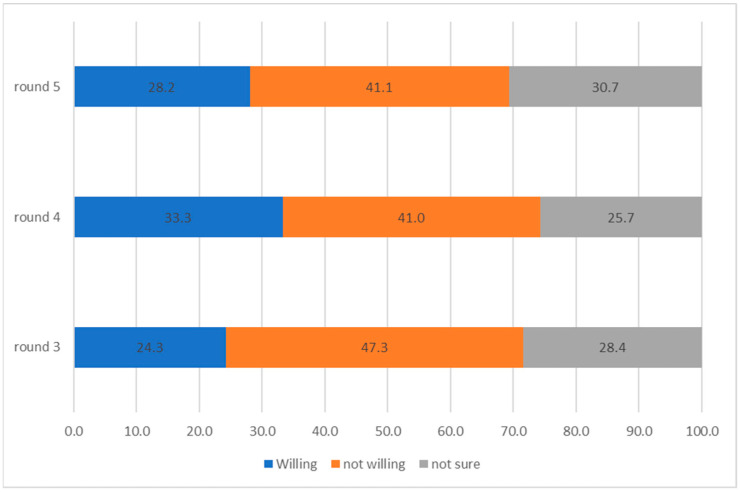
Vaccination status, over time, in %.

**Figure 2 vaccines-11-01272-f002:**
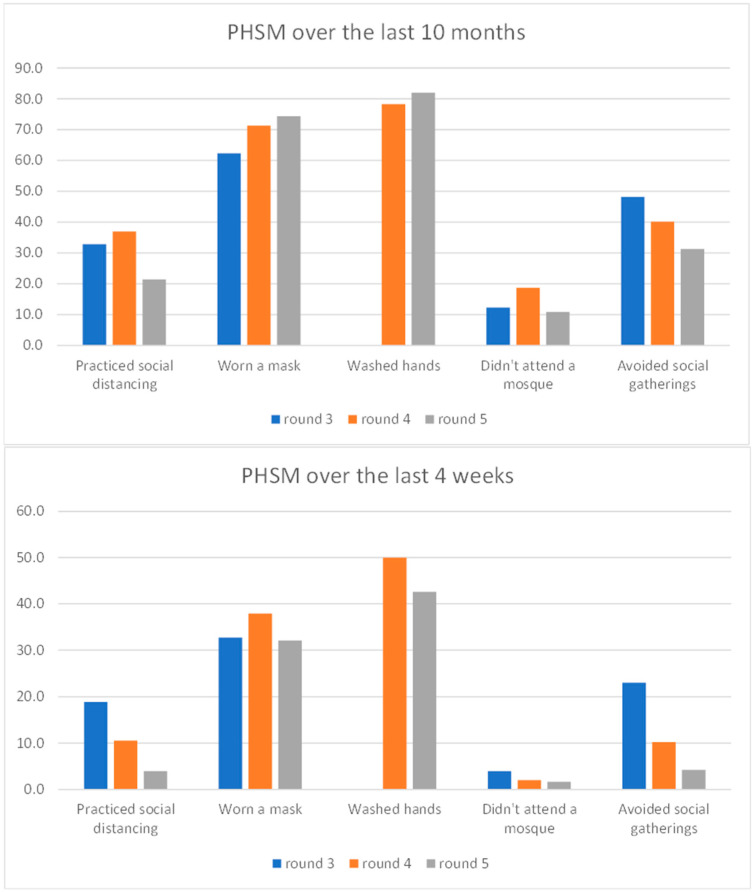
Selected PHSM, over time, in %.

**Table 1 vaccines-11-01272-t001:** Descriptive statistics of the data used in the analysis.

	Round 3 (March 2021)		Round 4 (Aug/Sept 2021)		Round 5 (April 2022)	
	%	Number	%	Number	%	Number
**Age**						
under 20	6.4	89	7.4	104	7.0	101
21 to 30	29.3	408	31.6	446	28.7	416
31 to 40	31.8	443	30.7	433	31.2	452
41 to 50	21.3	296	18.1	255	20.9	303
51 to 60	9.1	127	10.3	145	9.4	136
61 to 70	1.7	23	2.1	29	2.2	32
71 and above	0.4	6	0.0	0	0.7	10
**Education**						
cannot read and write	10.0	140	10.4	147	7.2	105
can read and write	18.2	255	15.6	220	15.6	226
basic	12.0	167	12.2	173	14.9	216
secondary	28.7	401	27.7	391	31.0	450
college degree	29.3	409	31.8	449	27.8	404
masters or PhD	1.9	26	2.3	33	3.5	50
**Gender**						
female	46.6	651	42.9	606	34.1	494
male	53.4	747	57.1	807	66.0	957
**Occupation**						
agricultural	9.0	126	10.5	149	8.9	129
educational	14.5	203	17.1	242	14.0	203
housewife	24.7	345	19.9	281	17.4	252
office	12.4	173	11.8	167	16.3	237
student	13.8	193	12.7	179	15.2	220
unemployed	6.9	97	5.9	84	4.1	60
handicraft	9.7	136	14.1	199	19.0	276
other	8.9	125	7.9	112	5.1	74
**Likely to become sick with COVID-19**						
I do not know	40.7	554	36.2	506	38.0	550
Yes	41.3	563	49.0	685	49.8	721
No	18.1	246	14.9	208	12.3	178
**Trust in the official information from the authorities**						
no confidence	26.8	337	16.5	210	14.6	193
little confidence	38.8	489	35.1	447	26.9	356
confident	27.4	345	36.1	460	50.7	670
total confidence	7.0	88	12.4	158	7.8	103
**PHSM in the last 10 months**						
practised social distancing	32.8	447	37.0	517	21.4	310
worn a mask	62.3	849	71.3	997	74.3	1077
washed hands	N/A	N/A	78.3	1095	82.0	1188
**PHSM in the last 4 weeks**						
practised social distancing	18.9	257	10.5	147	3.9	57
worn a mask	32.7	446	37.9	530	32.1	464
washed hands	N/A	N/A	50.0	699	42.6	617

Notes: PHSM—public health and social measures.

**Table 2 vaccines-11-01272-t002:** Round 5, vaccination status and socio-demographic characteristics.

	Willing		Not Sure		Not Willing		
	%	Number	%	Number	%	Number	chi2 *p*-Value
**Age**							
under 20	37	37	32	32	31	31	<0.001
21 to 30	33.4	139	30.8	128	35.8	149	
31 to 40	29.3	132	29.5	133	41.2	186	
41 to 50	21.5	65	32	97	46.5	141	
51 to 60	18.4	25	32.4	44	49.3	67	
61 to 70	18.8	6	25	8	56.3	18	
71 and above	30	3	30	3	40	4	
**Education**							
cannot read and write	14.3	15	18.1	19	67.6	71	<0.001
can read and write	18.1	41	29.2	66	52.7	119	
basic	34	73	29.8	64	36.3	78	
secondary	28.7	129	33.6	151	37.8	170	
college degree	31.3	126	32.5	131	36.2	146	
masters or PhD	48	24	28	14	24	12	
**Gender**							
female	19.5	96	29.9	147	50.6	249	<0.001
male	32.6	312	31.1	298	36.3	347	
**Occupation**							
agricultural	14	18	28.7	37	57.4	74	<0.001
educational	30.1	61	34	69	36	73	
housewife	16.7	42	28.7	72	54.6	137	
office	31.2	74	35	83	33.8	80	
student	38.8	85	31.1	68	30.1	66	
unemployed	20	12	38.3	23	41.7	25	
handicraft	30.8	85	27.5	76	41.7	115	
other	41.9	31	23	17	35.1	26	
**Likely to become sick with COVID-19**							
I do not know	20.9	115	33.6	185	45.5	250	<0.001
yes	36.8	265	30.9	223	32.3	233	
no	15.7	28	20.8	37	63.5	113	
**Public Health and Social Measures over the last 10 months**	Willing		Not sure		Not willing		
**Practised social distancing**	%	number	%	number	%	number	chi2 *p*-value
no	25.9	295	30.7	350	43.4	494	<0.001
yes	36.5	113	30.7	95	32.9	102	
**Worn a mask**							
no	13.2	49	25.3	94	61.6	229	<0.001
yes	33.3	359	32.6	351	34.1	367	
**Stayed away from the mosque**							
no	26.7	345	31	400	42.3	547	<0.001
yes	40.1	63	28.7	45	31.2	49	
**Wash hands**							
no	13.8	36	24.1	63	62.1	162	<0.001
yes	31.3	372	32.2	382	36.5	434	
**Avoided social gatherings**							
no	26.4	263	29.6	295	44	438	<0.001
yes	32	145	33.1	150	34.9	158	
**Public Health and Social Measures over the last 4 weeks**	Willing		Not sure		Not willing		
**Practised social distancing**	%	number	%	number	%	number	chi2 *p*-value
no	27.5	383	31	431	41.5	578	<0.001
yes	43.9	25	24.6	14	31.6	18	
**Worn a mask**							
no	21	207	30.7	302	48.3	476	<0.001
yes	43.3	201	30.8	143	25.9	120	
**Stayed away from the mosque**							
no	27.7	394	31	442	41.3	589	<0.001
yes	58.3	14	12.5	3	29.2	7	
**Wash hands**							
no	21.4	178	29.7	247	48.9	407	<0.001
yes	37.3	230	32.1	198	30.6	189	
**Avoided social gatherings**							
no	27.2	377	31	430	41.9	581	<0.001
yes	50.8	31	24.6	15	24.6	15	

**Table 3 vaccines-11-01272-t003:** Round 5, vaccination status and knowledge regarding COVID-19.

	Willing		Not Sure		Not Willing		
Knowledge to Protect Yourself from the Virus	%	Number	%	Number	%	Number	chi2 *p*-Value
no knowledge	2.3	1	16.3	7	81.4	35	<0.001
needs improvement	12.3	32	27.3	71	60.4	157	
good	32.3	265	33.7	277	34	279	
very good	28.9	54	34.8	65	36.4	68	
excellent	40.6	56	18.1	25	41.3	57	
**Trust in the official information from the authorities**							
no confidence	14	27	22.8	44	63.2	122	<0.001
little confidence	18.5	66	36.5	130	44.9	160	
confident	37.6	252	31.5	211	30.9	207	
total confidence	42.7	44	21.4	22	35.9	37	
**Trust in your own ability to deal with the virus**							
no confidence	19.5	25	29.7	38	50.8	65	<0.001
little confidence	16.3	54	33.5	111	50.2	166	
confident	34.4	226	31.2	205	34.4	226	
total confidence	41.2	70	27.1	46	31.8	54	
**How dangerous do you think the COVID-19 virus is**							
it is not dangerous	2.6	7	24.9	66	72.5	192	<0.001
more or less dangerous	29.5	189	33	211	37.5	240	
very dangerous	40.3	210	31.1	162	28.6	149	

**Table 4 vaccines-11-01272-t004:** Round 5, vaccination status and COVID-19 vaccine beliefs.

	Willing		Not Sure		Not Willing		
Vaccine Is Effective	%	Number	%	Number	%	Number	chi2 *p*-Value
no	10.3	65	37.7	238	52	328	<0.001
yes	48.2	339	25.8	181	26	183	
**Vaccine has side effects**							
no	48.1	317	22.9	151	29	191	<0.001
yes	12.9	87	39.7	268	47.7	320	

**Table 5 vaccines-11-01272-t005:** Round 5, vaccination status and most trusted COVID-19 information source.

Most Trusted Source	Willing		Not Sure		Not Willing		
	%	Number	%	Number	%	Number	chi2 *p*-Value
**TV**							
first mention	35.6	252	28.4	201	36	255	<0.001
second mention	15.8	15	39	37	45.3	43	
third mention	22.6	7	35.5	11	41.9	13	
**Radio**							
first mention	16.8	19	37.2	42	46	52	0.56
second mention	10.3	3	37.9	11	51.7	15	
third mention	28.6	4	21.4	3	50	7	
**Whatsapp**							
first mention	33.1	46	33.8	47	33.1	46	0.21
second mention	28.6	22	29.9	23	41.6	32	
third mention	14.7	5	35.3	12	50	17	
**Social media**							
first mention	36.4	51	34.3	48	29.3	41	0.21
second mention	37.1	39	31.4	33	31.4	33	
third mention	21.2	11	34.6	18	44.2	23	
**Communication materials**							
first mention	47	31	27.3	18	25.8	17	0.16
second mention	42.9	18	35.7	15	21.4	9	
third mention	20	5	40	10	40	10	
**Health unit**							
first mention	29.5	31	22.9	24	47.6	50	0.32
second mention	18.4	16	26.4	23	55.2	48	
third mention	30	12	30	12	40	16	
**Family**							
first mention	10.4	23	24.3	54	65.3	145	0.07
second mention	17.9	17	27.4	26	54.7	52	
third mention	23.7	9	29	11	47.4	18	
**Friends**							
first mention	28	30	22.4	24	49.5	53	0.03
second mention	20.6	21	24.5	25	54.9	56	
third mention	6.7	4	31.7	19	61.7	37	
**Community health workers**							
first mention	26.8	37	32.6	45	40.6	56	0.29
second mention	25.6	21	28.1	23	46.3	38	
third mention	21.3	13	21.3	13	57.4	35	
**Volunteers**							
first mention	39.5	43	23.9	26	36.7	40	0.35
second mention	27.1	19	24.3	17	48.6	34	
third mention	33.3	15	31.1	14	35.6	16	
**Community leaders**							
first mention	50	8	25	4	25	4	0.07
second mention	5.9	1	47.1	8	47.1	8	
third mention	24	6	32	8	44	11	
**Religious leaders**							
first mention	22.3	37	33.7	56	44	73	0.52
second mention	25.6	22	24.4	21	50	43	
third mention	23.2	19	25.6	21	51.2	42	
**Traditional healers**							
first mention	30.8	4	15.4	2	53.9	7	0.27
second mention	25	3	25	3	50	6	
third mention	0	0	45.5	5	54.6	6	
**A person from the community**							
first mention	44.4	4	22.2	2	33.3	3	0.2
second mention	66.7	6	0	0	33.3	3	
third mention	28.6	8	35.7	10	35.7	10	

## Data Availability

Data available upon request.

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
