# Peer review of "COVID-19 Vaccination Personas in Yemen: Insights from Three Rounds of a Cross-Sectional Survey"

_vaccines, 2023, doi:10.3390/vaccines11071272_

Round 1

Reviewer 1 Report

The authors have made an interesting attempt at “COVID-19 Vaccination Personas in Yemen: Insights from Three Rounds of a Cross-Sectional Survey.” The manuscript is interesting; however, the authors need to justify the scientific writing manuscript. Some of the general comments are provided below:

1.     Can you elaborate on the demographic and socioeconomic characteristics that were associated with willingness to vaccinate? Which specific variables within these categories were found to be significant correlates of vaccination status?

2.     How were the respondents selected for each round of the survey? Was the sampling method representative of the population in Yemen?

3.     Can you provide more information on the phone-based administration in the south and the face-to-face administration in the north? Were there any differences in response rates or data quality between the two methods?

4.     Considering that the sample is fairly young, does this age distribution reflect the overall population demographics in Yemen?

5.     Did the survey account for external factors that might have influenced changes in PHSM practices, such as changes in government guidelines, local COVID-19 transmission rates, or public perception of the pandemic?

6.     Did the article discuss potential reasons or underlying factors for the unwillingness to be vaccinated among this persona? Were alternative explanations explored?

7.     Did the analysis account for the temporal relationship between practicing PHSM and vaccine willingness? For example, did it consider whether individuals' lack of adherence to PHSM was influenced by their hesitancy towards vaccination or vice versa?

8.     Did the article discuss potential implications of relying on family and friends as the most trusted source of COVID-19 information for vaccine acceptance and uptake among this persona?

9.     Based on the survey findings, what specific recommendations or interventions can be proposed to address vaccine hesitancy and improve vaccination coverage in Yemen?

Author Response

Reviewer 1

The authors have made an interesting attempt at “COVID-19 Vaccination Personas in Yemen: Insights from Three Rounds of a Cross-Sectional Survey.” The manuscript is interesting; however, the authors need to justify the scientific writing manuscript. Some of the general comments are provided below:

Response: We thank the reviewer for the positive feedback. We further outline how we have dealt with different comments in the point-by-point response letter below.

  1. Can you elaborate on the demographic and socioeconomic characteristics that were associated with willingness to vaccinate? Which specific variables within these categories were found to be significant correlates of vaccination status?

Response: We thank the reviewer for this comment. We have further elaborated on this point in the results section of the paper. In addition, and as a complementary analysis, we have also conducted the standard logit modelling analysis, where the three vaccination personas appear as dependent variables in three separate models. The explanatory variables were grouped in three major groups: (i) demographic and socio-economic variables (e.g. age, gender); (ii) practicing of some of the most common public health and social measures (e.g. wearing a mask, washing hands); and (iii) beliefs about the Covid-19 vaccines (e.g. vaccines are effective, vaccines have side effects). The results are reported as Appendix tables (tabled A4-A6). The unequivocal result from this modelling exercise is that certain demographic characteristics (e.g. gender) as well as beliefs regarding the vaccines (effectiveness, side effects) are the statistically significant explanatory variables vis-à-vis the decision to obtain a Covid-19 vaccine.

Against this background and in the end of the results section we have added the following paragraph: “As a complementary analysis, we have also conducted the standard logit modelling analysis, where the three vaccination personas appear as dependent variables in three separate models. The explanatory variables were grouped in three major groups: (i) demographic and socio-economic variables (e.g. age, gender); (ii) practicing of some of the most common public health and social measures (e.g. wearing a mask, washing hands); and (iii) beliefs about the Covid-19 vaccines (e.g. vaccines are effective, vaccines have side effects). The results are reported as Appendix tables (tabled A4-A6). The analysis supports the findings from the descriptive stats in that demographic variables and variables capturing Covid-19 vaccines beliefs explain the decision to obtain a Covid-19 vaccination”.

  1. How were the respondents selected for each round of the survey? Was the sampling method representative of the population in Yemen?

Response: We thank the reviewer for this comment. The sample size was determined based on several factors, including a target population size of 14 million people (population at age >17), a confidence level of 95%, and a margin of error of approximately 2.5%. These figures were carefully considered to ensure that the sample size was both statistically significant and logistically feasible to implement.

As explained in the methodology section of the paper, the data gathering was done through face-to-face and phone-based interviews. This methodology for conducting interviews applied in this survey, thus, differs from online and email surveys in that it ensures to conduct the specified number of interviews for each round. In the event that a planned interview couldn’t be completed due to refusal or unavailability, simple substitution method was used by selecting the second nearest house or phone number for the interview to maintain the specified sample size. Ultimately, the data collection aimed to achieve the target of 1,400 interviews in each round of the survey.

The respondents were selected randomly as explained in the methodology part of the paper. The proportion of gender in the sample closely reflects that of the actual population in Yemen. However, for other demographic variables such as education and occupation, accurate and up-to-date data on the Yemeni population is not available, and therefore, the survey could not be considered representative across these additional dimensions; subsequently, no survey weights were derived that would ensure representativeness of the data. While some estimation and projection of some demographic data exist, it is not reliable enough to allow for accurate weighting of the survey results.

Against this background, we have added the following couple of paragraphs to the methodology section of the paper: “The sample size was determined based on several factors, including a target population size of 14 million people (population at age >17), a confidence level of 95%, and a margin of error of approximately 2.5%”.

When it comes to the representativeness of the data, we have added the following sentence to the limitations section of the paper: “As the estimation and projection of demographic data in Yemen is of poor quality, the survey did not develop survey weights. More specifically, the results are not weighted for survey weights to address representativeness of the sample.”

  1. Can you provide more information on the phone-based administration in the south and the face-to-face administration in the north? Were there any differences in response rates or data quality between the two methods?

Response: We thank the reviewer for this comment. Regarding the data collection specifically, as outlined in the paper, in the North, the interviews were done face-to-face. For selection of the enumeration areas, governorates were identified to serve as the Primary Sampling Units (PSUs); from there, governorates were implicitly stratified to randomly select clusters while also considering the ease of access (no restriction from authorities, not significantly affected by conflict etc.). Selection of individuals to be interviewed in each governorate was based on simple random selection. The enumerator identified the nearest clear landmark in the centre of the specified location and then the enumerator selected the houses located on their right hand according to the specified counting distance between the samples (the fifth house). In case of rejection or closed household, a simple substitution method was used, and the second nearest house was selected (the sixth house); and the enumerator continues from there.

In the south and due to movement restrictions during COVID-19, the interviews were done through phone. Random numbers were selected from a dataset of phone numbers in the south using excel RAND function. This method is less representative as the dataset does not include all phone numbers in the south. Therefore, when COVID-19 restrictions started to ease up in the south, most of the interviews (84%) in Round 5 were done face-to-face.

On response rate, the number of interviews conducted are equal to the specified sample size in each round, both in the South and in the North.

Against this background, we added the following paragraph to the methods section of the paper: “In the North of the country, for the selection of the enumeration areas, governorates were identified to serve as Primary Sampling Units (PSUs); based on this, governorates were implicitly stratified to allow for a random selection of clusters while considering the ease of access during the selection (e.g. not a conflict affected zone, no restrictions from authorities). In turn, a simple random selection was applied for selection to be interviewed in each governorate. By contrast, the interviews in the South were done over the phone. More specifically, random numbers were selected from a dataset of phone numbers in the South (noting that this method impacts upon the representativeness of the sample in the South). The application of these different data collection methods did not significantly impact upon the response rate across the country. In other words, the number of interviews conducted are equal to the specified sample size in each round, both in the South and in the North”. 

  1. Considering that the sample is fairly young, does this age distribution reflect the overall population demographics in Yemen?

Response: We thank the reviewer for this comment. The respondents were selected randomly as explained in the methodology part of the paper. The proportion of gender in the sample closely reflects that of the actual population in Yemen. However, for other demographic variables such as education and occupation, accurate and up-to-date data on the Yemeni population is not available, and therefore, the survey could not be considered representative across these additional dimensions; subsequently, no survey weights were derived that would ensure representativeness of the data. While some estimation and projection of some demographic data exist, it is not reliable enough to allow for accurate weighting of the survey results.

As further explained in one of the responses to the previous query above, we have added the following paragraph to the limitations section of the paper: “As the estimation and projection of demographic data in Yemen is of poor quality, the survey did not develop survey weights. More specifically, the results are not weighted for survey weights to address representativeness of the sample”.

  1. Did the survey account for external factors that might have influenced changes in PHSM practices, such as changes in government guidelines, local COVID-19 transmission rates, or public perception of the pandemic?

Response: We thank the reviewer for this comment. We have exhaustively used all of the questions available in the survey when doing the analysis. With that said, as the survey is a rapid KAP (knowledge, attitudes, practices) survey, it did not account for any additional external factors that may impact upon PHSM practices, such as changes in government guidelines, local COVID-19 transmission rates or the public perception of the pandemic.

  1. Did the article discuss potential reasons or underlying factors for the unwillingness to be vaccinated among this persona? Were alternative explanations explored?

Response: We thank the reviewer for this comment. As indicated by the additional analysis we have conducted (i.e. logit modelling analysis), gender was one of the main demographic variables capturing this persona (with females being more likely to be unwilling to receive the vaccine relative to males). There have been a few reasons for this finding. Part of the explanation why male gender has only sometimes been associated with higher COVID‑19 vaccination intent when comparing results from around the world may be differences in the social and cultural characteristics of the population studied; for example, in countries where men spend more time than women out of the house for both work and nonwork‑related activities, men may be more concerned about their risk of contracting the virus causing COVID‑19 disease and therefore more eager for vaccination. Furthermore, in countries where men have greater access to health care services and the means to pay for vaccination than women, men may be more interested in COVID‑19 vaccination (Dube et al, 2014; Wagner et al, 2019). In addition, it has been argued that women are more likely to embrace the conspiracy beliefs about the virus (Abu-Farha et al, 2021). Other potential factors that can contribute to higher rates of vaccine hesitancy among females include the higher levels of fear of injections or side effects, and the observation that the disease is more deadly in males (Abu-Farha et al, 2021).

Against this background, we have added the following paragraph to the discussion section in the paper: “Part of the explanation why male gender has only sometimes been associated with higher COVID‑19 vaccination intent when comparing results from around the world may be differences in the social and cultural characteristics of the population studied; for example, in countries where men spend more time than women out of the house for both work and nonwork‑related activities, men may be more concerned about their risk of contracting the virus causing COVID‑19 disease and therefore more eager for vaccination. Furthermore, in countries where men have greater access to health care services and the means to pay for vaccination than women, men may be more interested in COVID‑19 vaccination (Dube et al, 2014; Wagner et al, 2019). In addition, it has been argued that women are more likely to embrace the conspiracy beliefs about the virus (Abu-Farha et al, 2021). Other potential factors that can contribute to higher rates of vaccine hesitancy among females include the higher levels of fear of injections or side effects, and the observation that the disease is more deadly in males (Abu-Farha et al, 2021).”

Referenced papers:

  1. Dubé E, Gagnon D, Nickels E, Jeram S, Schuster M. Mapping vaccine hesitancy—Country‑specific characteristics of a global phenomenon. Vaccine 2014;32:6649‑
  2. Wagner AL, Masters NB, Domek GJ, Mathew JL, Sun X, Asturias EJ, et al. Comparisons of vaccine hesitancy across five low‑and middle‑income countries. Vaccines 2019;7:155.
  3. Abu-Farha R, Mukattash T, Itani R, Karout S, Khojah HMJ, Abed Al-Mahmood A, Alzoubi KH. Willingness of Middle Eastern public to receive COVID-19 vaccines. Saudi Pharm J. 2021 Jul;29(7):734-739. Doi: 10.1016/j.jsps.2021.05.005. Epub 2021 May 31. PMID: 34093059; PMCID: PMC8165039.

  1. Did the analysis account for the temporal relationship between practicing PHSM and vaccine willingness? For example, did it consider whether individuals’ lack of adherence to PHSM was influenced by their hesitancy towards vaccination or vice versa?

Response: We thank the reviewer for this comment. In order to capture this relationship, we have pooled the three waves together and have used the three vaccination personas as dependent variables in three separate bivariate logit models (where the variables capturing different PHSM are used as independent variables). We have repeated the analysis twice, first using PHSM practiced over the last 10 months and then over the last four weeks. The models also control for the survey wave (i.e. taking into account any temporal changes occurring over the three different waves). The findings (reported in the Appendix Tables A7-A8) are unequivocal: those more willing to vaccinate are also more willing to adhere to various PHSM (both, over the last 10 months as well as over the last 4 weeks).

Against this background, we have added the following paragraph to the results section: “In order to capture the PHSM/vaccination status nexus over time, we have pooled the three waves together and have used the three vaccination personas as dependent variables in three separate bivariate logit models (where the variables capturing different PHSM are used as independent variables). We have repeated the analysis twice, first using PHSM practiced over the last 10 months and then over the last four weeks. The models also control for the survey wave (i.e. taking into account any temporal changes occurring over the three different waves). The findings (reported in the Appendix Tables A7-A8) are unequivocal: those more willing to vaccinate are also more willing to adhere to various PHSM (both, over the last 10 months as well as over the last 4 weeks)”.

  1. Did the article discuss potential implications of relying on family and friends as the most trusted source of COVID-19 information for vaccine acceptance and uptake among this persona?

Response: Relying on family and friends is an important source of Covid-19 vaccine information among this type of persona. Thus, identifying community/larger family champions that could help explain how the vaccines work could go long way in alloying some of the fears associated with Covid-19 vaccine fears. Against this background, in the recommendation section, we added the following sentence regarding the use of this information channel: “Identifying vaccine champions among families/communities could further alloy some of the fears associated with vaccines (e.g. fears of side effects).”

  1. Based on the survey findings, what specific recommendations or interventions can be proposed to address vaccine hesitancy and improve vaccination coverage in Yemen?

Response: There are a few policy recommendations that could be applied in this case, based on the findings from this survey. As we indicate in the discussion section, a deeper engagement with the communities (e.g. via outreach healthcare workers and/or religious leaders) could help convey some of the messages regarding Covid-19 vaccines concerns (e.g. fear of side effects) ultimately reducing vaccine hesitancy.

Thus, at the end of the paper, the following paragraph captures the specific recommendations that could be applied in the case of Yemen: “Finally, there are some broad policy recommendations that stem from this research efforts. Any focus on individual motivation for vaccination relies on the basic requirement that adequate vaccination services are made available to all communities. That said, outreach to communities and a localised focus on the needs of those who are undecided about vaccination can be effective in increasing uptake, thereby also increasing the social norm around being vaccinated. Supplying them with information about the COVID-19 vaccines (e.g. safety, effectiveness, side effects) and access to trusted and skilled health workers could help in allaying their fears and increase confidence in the vaccines. Religious leaders and other community leaders (including females) can have a strong influence on communities in Yemen, both positive and negative – and also should be considered key partners, especially in terms of understanding and addressing the needs of local communities”. 

Reviewer 2 Report

Introduction: 

This study report is unique and important because of the devastating situation in Yemen.

Methods:

The methods section is very brief. There was not enough information for readers to evaluate the representativeness of the survey sample. If the sample was not representative, the conclusions would be biased. At least, the following information has to be reported:

  1. How and why did the survey select 1400 participants?
  2. Were same respondents selected over the 5 rounds of survey?
  3. What were the response rates? 
  4. Some evidence regarding the representativeness of the 1400 respondents to the population in Yemen.

Readers also need to know more about the validity and reliability of the instrument. For example, how were the questions framed in the questionnaire? For example, How did the knowledge of symptoms, transmission, and prevention; (ii) sources of information; (iii) risk perception; (iv) information needs; (v) COVID-related stigma; and (vi) hesitancy or acceptance of COVID-19 vaccine were measured. At least, the authors should tell reader where they can find the questionnaire.

The results: 

  1. The contents in the tables and text should be complementary to each other. Same information do not have to be repeated in both the table and text
  2. Tables and charts should be self-explanatory. Acronyms should be noted at the bottom of tables/charts. 
  3. It is easy to get a p-value below 0.05 for multiple-category comparisons in Chi-square tests. No insights would be gained from this kind of Chi-square analyses.

Discussion. 

It might be true that, as it's claimed by the authors, this study is the first "comprehensive attempt" to understand the COVID vaccination willingness among Yemenis. However, the meaning of vaccine hesitancy in a society with vaccines widely available and affordable is different from a society where vaccines are in short supply. The authors might need to put the study findings under the context of Yemen. 

Author Response

Reviewer 2

Introduction: 

This study report is unique and important because of the devastating situation in Yemen.

Response: We thank the reviewer for the positive feedback. In the text below, we further outline, in a point-by-point basis how we have dealt with the comments proposed by the reviewer. 

Methods: The methods section is very brief. There was not enough information for readers to evaluate the representativeness of the survey sample. If the sample was not representative, the conclusions would be biased. At least, the following information has to be reported:

Response: We thank the reviewer for the positive feedback. In the text below, we further outline, in a point-by-point basis how we have dealt with the comments proposed by the reviewer.

How and why did the survey select 1400 participants?

Response: We thank the reviewer for this comment. The sample size was determined based on several factors, including a target population size of 14 million people (population at age >17), a confidence level of 95%, and a margin of error of approximately 2.5%. These figures were carefully considered to ensure that the sample size was both statistically significant and logistically feasible to implement.

Against this background, we have added the following sentence to the methods section: “The sample size was determined based on several factors, including a target population size of 14 million people (population at age >17), a confidence level of 95%, and a margin of error of approximately 2.5%.”

Were same respondents selected over the 5 rounds of survey?

Response: We thank the reviewer for this comment. The sampling method used over the rounds is the same: simple random sampling for selecting respondents from the same governorates. However, we did not aim to target the same respondents over the three separate rounds and we did not check to see if same respondents were the same as we did not record personal information of the respondents.

Against this background, we have added the following sentence to the methods section: “The three rounds of the survey followed a repeated cross-section format (rather than a longitudinal survey format); thus the same individuals do not appear in all three rounds of the survey”. 

What were the response rates?

Response: We thank the reviewer for this comment. As explained in the methodology section of the paper, the data gathering was done through face-to-face and phone-based interviews. This methodology for conducting interviews applied in this survey, thus, differs from online and email surveys in that it ensures to conduct the specified number of interviews for each round. In the event that a planned interview cannot be completed due to refusal or unavailability, simple substitution method was used by selecting the second nearest house or phone number for the interview to maintain the specified sample size. Ultimately, the data collection aimed to achieve the target of 1400 interviews in each round of the survey. 

Some evidence regarding the representativeness of the 1400 respondents to the population in Yemen.
Response:  We thank the reviewer for this comment. The respondents were selected randomly as explained in the methodology part of the paper. The proportion of gender in the sample closely reflects that of the actual population in Yemen. However, for other demographic variables such as education and occupation, accurate and up-to-date data on the Yemeni population is not available, and therefore, the survey could not be considered representative across these additional dimensions; subsequently, no survey weights were derived that would ensure representativeness of the data. While some estimation and projection of some demographic data exist, it is not reliable enough to allow for accurate weighting of the survey results.

Against this background, we have added the following sentence to the limitation section of the paper: “As the estimation and projection of demographic data in Yemen is of poor quality, the survey did not develop survey weights. More specifically, the results are not weighted for survey weights to address representativeness of the sample”.

Readers also need to know more about the validity and reliability of the instrument. For example, how were the questions framed in the questionnaire? For example, How did the knowledge of symptoms, transmission, and prevention; (ii) sources of information; (iii) risk perception; (iv) information needs; (v) COVID-related stigma; and (vi) hesitancy or acceptance of COVID-19 vaccine were measured. At least, the authors should tell reader where they can find the questionnaire.

Response: We thank the reviewer for this comment. The questionnaire underwent a thorough review process, with input from several partners and counterparts, including WHO as well as members of various coordination and decision-making bodies such as the COVID-19 task force and the Risk Communication Community Engagement working group. Additionally, the questionnaire was pre-tested with selected participants to ensure clarity and relevance. The questionnaire is not available in the public domain and it is available upon request from the UNICEF Country Office in Yemen. 

Against this background, we have added the following paragraph to the methods section: “The questionnaire underwent a thorough review process, with input from several partners and counterparts, including WHO as well as members of various coordination and decision-making bodies such as the COVID-19 task force and the Risk Communication Community Engagement working group. Additionally, the questionnaire was pre-tested with selected participants to ensure clarity and relevance. The questionnaire is not available in the public domain and it is available upon request from the UNICEF Country Office in Yemen.”

The results: 

The contents in the tables and text should be complementary to each other. Same information do not have to be repeated in both the table and text

Response: We thank the reviewer for this comment. We have re-drafted some of the results section in order not to repeat the same information twice. By doing the redrafting, in the amended version of the manuscript, the text provides a commentary on the results without heavily repeating what is already reported in the tables. 

Tables and charts should be self-explanatory. Acronyms should be noted at the bottom of tables/charts. 

Response: We thank the reviewer for this comment. We have amended the tables and they now, where applicable, also include explanation of some of the abbreviations that have been used (e.g. PHSM – public health and social measures). 

It is easy to get a p-value below 0.05 for multiple-category comparisons in Chi-square tests. No insights would be gained from this kind of Chi-square analyses.
Response: We thank the reviewer for this comment. As a complementary analysis, we have also conducted the standard logit modelling analysis, where the three vaccination personas appear as dependent variables in three separate logit models. The explanatory variables were grouped in three major groups: (i) demographic and socio-economic variables (e.g. age, gender); (ii) practicing of some of the most common public health and social measures (e.g. wearing a mask, washing hands); and (iii) beliefs about the Covid-19 vaccines (e.g. vaccines are effective, vaccines have side effects). The results are reported as Appendix tables (tabled A4-A6). The unequivocal results from this modelling attempt is that certain demographic characteristics (e.g. gender) as well as beliefs regarding the vaccines (effectiveness, side effects) are the most common explanatory variables vis-à-vis the decision to obtain a Covid-19 vaccine. 

Against this background and in the end of the results section we have added the following paragraph: “As a complementary analysis, we have also conducted the standard logit modelling analysis, where the three vaccination personas appear as dependent variables in three separate models. The explanatory variables were grouped in three major groups: (i) demographic and socio-economic variables (e.g. age, gender); (ii) practicing of some of the most common public health and social measures (e.g. wearing a mask, washing hands); and (iii) beliefs about the Covid-19 vaccines (e.g. vaccines are effective, vaccines have side effects). The results are reported as Appendix tables (tabled A4-A6). The analysis supports the findings from the descriptive stats in that demographic variables and variables capturing Covid-19 vaccines beliefs explain the decision to obtain a Covid-19 vaccination”. 

Discussion.  

It might be true that, as it's claimed by the authors, `this study is the first "comprehensive attempt" to understand the COVID vaccination willingness among Yemenis. However, the meaning of vaccine hesitancy in a society with vaccines widely available and affordable is different from a society where vaccines are in short supply. The authors might need to put the study findings under the context of Yemen. 

Response: We thank the reviewer for this comment and we agree with the reviewer that the results need to be put into Yemeni context. In the discussion section, we have added a paragraph where specific aspects of vaccine hesitancy for Yemen are considered (also building on available evidence from the literature). In addition, we have added the following couple of sentences in the discussion section of the paper: “These findings need to be interpreted within the broader context of the political situation in Yemen, which affected the availability of accurate information and of vaccination services (including availability of vaccines), particularly in the northern DFA controlled provinces”.

Round 2

Reviewer 1 Report

The authors have addressed my queries and the manuscript is acceptable for publication. 

Reviewer 2 Report

The authors have addressed my concerns in the revised manuscript. I have no more comments.